# Phase Angle as a Risk Factor for Mortality in Patients Undergoing Peritoneal Dialysis

**DOI:** 10.3390/nu15234991

**Published:** 2023-12-01

**Authors:** Seok-Hui Kang, Jun-Young Do

**Affiliations:** Division of Nephrology, Department of Internal Medicine, College of Medicine, Yeungnam University, Daegu 42415, Republic of Korea; kangkang@ynu.ac.kr

**Keywords:** peritoneal dialysis, phase angle, patient survival, technique survival

## Abstract

Phase angle (PhA) is measured using bioimpedance analysis and calculated using body reactance and resistance in the waveform at 50 kHz. Further studies are necessary to clarify the predictive efficacy of PhA in the mortality of peritoneal dialysis (PD) patients. The objective of this study was to assess the utility of PhA for predicting patient mortality and technique failure and compare the predictability of PhA with other risk factors. Our study had a retrospective cohort design. Our center routinely evaluates bioimpedance measurements for all prevalent PD patients (*n* = 199). The PhA was measured using multifrequency bioimpedance analysis. Our study evaluated patient and technique survival. There were 66, 68, and 65 patients in the low, middle, and high tertiles of PhA, respectively. The PhA values of the low, middle, and high tertiles were 3.6° (3.4–3.9), 4.4° (4.2–4.7), and 5.5° (5.2–6.0), respectively. The 5-year patient survival rates for the high, middle, and low tertiles were 100%, 81.7%, 69.9%, respectively (*p* < 0.001). The 5 year technique survival rates for the high, middle, and low tertiles were 91.9%, 74.8%, 63.7%, respectively (*p* = 0.004). Patient and technique survival increased as the PhA tertiles increased. Both univariate and multivariate Cox regression analyses demonstrated a consistent pattern. The prediction of patient or technique survival was better in PhA than in the other classical indicators. The present study demonstrated that PhA may be an effective indicator for predicting patient or technique survival in PD patients. Furthermore, it suggests that routine measurement of PhA and pre-emptive intervention to recover PhA according to causes of low PhA may help improve patient or technique survival in PD patients.

## 1. Introduction

As the global population continues to age and the prevalence of numerous comorbidities increases, chronic kidney disease has become a growing health concern [1]. The final stage of chronic kidney disease, known as kidney failure, necessitates one of three forms of kidney replacement therapies: hemodialysis, peritoneal dialysis (PD), or kidney transplantation [2]. Although the prevalence of PD is decreasing compared with that of hemodialysis or kidney transplantation, PD, performed using the patient’s peritoneal membrane, is an effective kidney replacement therapy with some benefits [3]. PD aids in the favorable removal of middle molecules and the stable removal of water compared with hemodialysis. However, despite these benefits, PD patients have a higher mortality than the general population [4]. A meta-analysis showed that various factors, including age, diagnosis of diabetes mellitus, cardiovascular disease, or peritonitis, the abnormal levels of serum albumin, hemoglobin, potassium, uric acid, and alkaline phosphatase, were classically associated with mortality in PD patients [5]. However, identifying new risk factors beyond these classic factors would help understand high mortality and reduce mortality rates through proper intervention.

Phase angle (PhA) is calculated using body reactance and resistance in the waveform at 50 kHz and is an indicator for predicting the health of the cell membrane [6]. Various indices, such as muscle mass or water content beyond simple PhA value, can also be measured using bioimpedance; however, these data are derived from linear regression equations using raw data. Therefore, PhA is increasingly recognized as a prognostic indicator without transformation from various equations. Previous studies have established associations between PhA and sarcopenia, nutritional status, vascular calcification, or arterial stiffness in individuals with chronic kidney disease [6,7,8]. In addition, a positive association between PhA and mortality as a hard outcome in PD patients has also been reported; however, further studies are necessary to clarify the predictive efficacy of PhA in the mortality of PD patients [9,10,11]. The objective of this study was to assess the utility of PhA for predicting patient mortality and technique failure and compare the predictability of PhA with other risk factors.

## 2. Materials and Methods

### 2.1. Study Population

This study adopted a retrospective cohort design. Bioimpedance measurements are routinely conducted for all prevalent PD patients at our center. Data were gathered from 214 prevalent PD patients who attended our hospital between September 2017 and November 2020. Among these, nine patients were excluded due to insufficient data, and the PhA of six patients was not evaluated due to an amputated limb. Consequently, 199 prevalent PD patients were included in the final analyses. Ethical approval for this study was obtained from the institutional review board of a local medical center, and the study was conducted in accordance with the principles of the World Medical Association Declaration of Helsinki (approval no: YUMC-2021-01-019, accessed on 15 January 2021).

### 2.2. Baseline Characteristics

We obtained baseline characteristics during a routine peritoneal membrane equilibration test while conducting the bioimpedance measurements. Baseline data included sex, age, use of automated PD, Davies comorbidity index, dialysis vintage (months), body mass index (kg/m^2^), weekly Kt/V_urea_, urine volume (mL/day), dialysate-to-creatinine concentration 4 h (DP4Cr) ratio, phosphorus (mg/dL), serum calcium (mg/dL), sodium (mEq/L), potassium (mEq/L), serum albumin (g/dL), C-reactive protein (mg/dL), alkaline phosphatase (IU/L), normalized protein equivalent of total nitrogen appearance (nPNA, g/kg/day), and intact parathyroid hormone (pg/mL) levels.

All laboratory investigations were conducted following an overnight fasting period. The Davies comorbidity index was determined based on a prior publication [12]. Briefly, this index incorporates comorbidities such as malignancy, peripheral vascular disease, ischemic heart disease, diabetes mellitus, left ventricular dysfunction, systemic collagen vascular disease, or other significant illnesses. Patients were classified as having low, intermediate, or high risk based on the number of comorbidities present: 0, 1–2, and ≥3, respectively. The DP4Cr ratio was measured using a modified 4.25% peritoneal equilibration test, calculated by comparing the creatinine level in dialysate 4 h after injection to blood creatinine levels. Weekly Kt/V_urea_ and nPNA were computed based on 24 h urine and dialysate collections, as previously described [13].

### 2.3. Assessment of PhA and Patient or Technique Survivals

PhA was determined using a multifrequency bioimpedance analysis system (InBody770, Seoul, Republic of Korea). PhA values were calculated based on the phase difference between the voltage and current waveforms at 50 kHz. Eight electrodes were placed on each patients’ hands and feet while they stood upright. To estimate PhA, the reactance (Xc) and resistance (R) values measured at 50 kHz were used in the following formula: PhA (°) = arctangent (Xc/R) × (180/π). Notably, the presence of peritoneal dialysate is associated with an underestimation of PhA [14]. To account for this effect, all PhA measurements were conducted after the abdomen had been emptied.

Our study evaluated patient and technique survival. All patients were asked to visit for follow up in July 2022. Patient death was defined as any death that occurred during the follow-up period, regardless of the cause. Patients who underwent kidney transplantation, switched to hemodialysis for at least 90 consecutive days, discontinued dialysis because of renal recovery, were lost to follow up, or transferred to another hospital were considered censored. Technique failure was defined as patient death of switch from PD to hemodialysis for at least 90 consecutive days [15]. If the patient had kidney transplantation, transferred to hemodialysis due to patient’s request without medical problem, discontinued dialysis owing to improved renal function, was lost to follow up, or transferred to another hospital, the data were categorized as censored.

### 2.4. Statistical Analyses

Statistical analysis was conducted using SAS software (version 9.4; SAS, Cary, NC, USA). Categorical variables were summarized as frequency counts and percentages and compared using Pearson’s chi-square or Fisher’s exact tests, depending on the sample size. Continuous variables were evaluated for normality using the Kolmogorov–Smirnov test. Normally distributed continuous variables are presented as mean ± standard deviation, while non-normally distributed variables are presented as median (interquartile range). Statistical comparisons of continuous variables were performed using the Kruskal–Wallis test for non-normally distributed data and one-way ANOVA for normally distributed data. Bonferroni’s post hoc test was employed for pairwise comparisons between the two groups.

We used Kaplan–Meier analysis to generate survival curves for each group and the Log-rank test to evaluate statistical significance. Cox regression analysis was used to estimate survival probabilities. For multivariate Cox regression analyses, we used the enter mode, and the covariates were selected for variables with statistical significance (*p* < 0.05) in univariate analyses. All variables satisfied the proportional hazard assumption. We assessed the variance inflation factors (VIF) using regression analysis with multivariate analysis variables (patient death as dependent variable and PhA, age, serum albumin, and Davies risk index as independent variables). The area under the receiver operating characteristic curve (AUROC) was used to assess the accuracy of a classifier in predicting mortality or technique failure. DeLong’s model was used to compare the AUROCs. In addition to traditional methods, we employed category-free integrated discrimination improvement (IDI) and net reclassification improvement (NRI) to assess model performance, adhering to the methodology outlined by Pencina et al. [16,17]. Pencina et al. [16,17] proposed the IDI and NRI methods to assess the improvement in predictive performance when a new factor is added to existing prediction models. Following their recommendations, we conducted IDI and NRI analyses to determine whether including PhA in a mortality prediction model based on existing risk factors enhances predictive accuracy. A *p*-value < 0.05 was considered statistically significant.

## 3. Results

### 3.1. Participants’ Clinical Characteristics

There were 66, 68, and 65 patients in the low, middle, and high tertiles of PhA, respectively (Table 1).

The PhA values of the low, middle, and high tertiles were 3.6° (3.4–3.9), 4.4° (4.2–4.7), and 5.5° (5.2–6.0), respectively. The patients in the high tertile were younger than those in the other tertiles. The urine volume was greater in patients in the high tertile than those in the other tertiles. The proportion of patients with a low risk of Davies comorbidity index and male sex was greater in the high tertile than those in the other tertiles. The serum albumin level increased as the tertile increased. No significant differences were observed in dialysis modality, C-reactive protein, dialysis vintage, DP4Cr, weekly Kt/V_urea_, serum phosphorus, calcium, sodium, potassium, intact parathyroid hormone, and alkaline phosphatase among the three tertiles.

### 3.2. Patient or Technique Survival according to PhA Tertiles

The survival curves for each group are presented in Figure 1.

Patient and technique survival increased as the PhA tertiles increased. Table 2 shows the results of Cox regression analyses for predicting patient death or technique failure.

Univariate Cox regression analyses revealed that PhA tertiles, age, serum albumin, and Davies comorbidity index were associated with patient death and technique failure. We evaluated the multicollinearity using regression analysis. The VIF was 1.16, 1.18, 1.35, and 1.18 for PhA, age, serum albumin, and Davies risk index, respectively. This indicated no multicollinearity among the independent variables of multivariate analyses. Multivariate analyses showed that a decrease in PhA tertile was independently associated with patient death. In addition, the decrease in PhA tertile had a trend of a higher hazard ratio (HR) for technique failure, but no statistical significance was observed.

Furthermore, we performed Cox regression analysis using raw PhA values. In univariate analysis, the HR (95% confidence interval[CI]) for patient mortality was 0.34 (0.22–0.53, *p* < 0.001) for every 1° increase in PhA value. In multivariate analysis adjusted for age, serum albumin, and Davies risk index, the HR was 0.40 (0.24–0.68, *p* = 0.001) for every 1° increase in PhA value. Additionally, in univariate analysis, the HR (95% CI) for technique failure was 0.53 (0.38–0.75, *p* < 0.001) for every 1° increase in PhA value. In multivariate analysis adjusted for age, serum albumin, and Davies risk index, the HR was 0.68 (0.46–1.01, *p* = 0.058) for every 1° increase in PhA value.

Eighteen patients in the low tertile and eleven in the middle tertile died during follow-up. In patients in the low tertile, the causes of death were cardiovascular disease for eight (44.4%), infection for nine (50.0%), and cachexia for one (5.6%). In patients in the middle tertile, the causes of death were cardiovascular disease for four (36.4%), infection for four (36.4%), cerebral hemorrhage for one (9.1%), and malignancy for two (18.2%). The numbers of technique failures in the low, middle, and high tertiles were 23, 16, and 7, respectively. In patients in the low tertile, the causes of technique failure were patient death for 18 (78.3%) and PD peritonitis for 5 (21.7%). In patients in the middle tertile, the causes of technique failure were patient death for eleven (68.7%), PD peritonitis for two (12.4%), uremic symptom for one (6.3%), PD catheter malfunction for one (6.3%), and malignancy for one (6.3%). In patients in the high tertile, the causes of technique failure were PD peritonitis for five (71.4%), tunnel infection for one (14.3%), and PD catheter malfunction for one (14.3%).

To evaluate the incremental predictive power of PhA in predicting patient death, we compared the probabilities of patient death and survival for models with and without PhA using relative IDI and category-free NRI (Table 3).

In patient death, the areas under the curves (AUCs) in models without or with PhA were 0.79 and 0.85, respectively. Models with PhA had better AUCs than those without PhA. Results from relative IDI and category-free NRI showed similar trends to those from a comparison between AUCs. In technique failure, AUCs in models without or with PhA were 0.71 and 0.73, respectively. Models with PhA had better AUCs than those without PhA, but statistical significance was not obtained.

### 3.3. The Comparison of Patient or Technique Survival among Various Indicators

The AUROC of the indicators for patient death or technique failure at the end point of follow-up is shown in Figure 2.

ROC analyses showed that PhA predicted patient death more efficiently than other indicators (AUCs were 0.80 [95% confidence interval (CI), 0.73–0.85] for PhA, 0.69 [0.62–0.76] for serum albumin, 0.56 [0.49–0.63] for BMI, 0.68 [0.60–0.74] for CRP, 0.58 [0.51–0.66] for nPNA, and 0.58 [0.51–0.66] for urine volume). Furthermore, PhA more effectively predicted technique failure than other indicators (AUCs were 0.69 [0.62–0.76] for PhA, 0.65 [0.58–0.72] for serum albumin, 0.54 [0.46–0.61] for BMI, 0.64 [0.57–0.71] for CRP, 0.54 [0.46–0.61] for nPNA, and 0.58 [0.50–0.65] for urine volume).

## 4. Discussion

We included prevalent PD patients and evaluated the association of patient or technique survival with PhA using Cox regression and AUROC analyses. Our study showed that PhA tertiles were associated with patient or technique survival. Univariate and multivariate Cox regression analyses showed similar trends. The prediction of patient or technique survival was better in PhA than in the other classical indicators.

PhA expresses the electrical function of cell membranes, and a low level of PhA is associated with cell death or reduced cell function [18]. Although this value has been classically used in calculating body compositions including bone, muscle mass, or fat mass, the raw PhA value is mainly influenced by extracellular and intracellular status, and these changes are associated with various pathologic conditions. Some studies investigated the association between PhA and nutritional status [8,9,10,11,19,20,21,22,23,24]. These can be connected with muscle status, physical functioning, and quality of life [25,26]. These changes can lead to an increased mortality rate.

Previous studies evaluated associations between PhA and nutrition status using composite or non-composite nutritional indicators in dialysis patients [8,9,10,11,19,20,21,22,23,24]. Sarmento-Dias et al. evaluated 61 stable PD patients and showed a positive association between PhA and arterial stiffness or vascular calcification using the simplified calcification score [7]. A study of Chinese PD patients showed a strong association between PhA and pulse wave velocity [24]. Studies regarding the association between PhA and cardiovascular risk factors, nutrition, or sarcopenia have been published in various populations; however, there are especially fewer studies regarding the association between PhA and survival as a hard outcome in PD patients than in those with hemodialysis or non-dialysis chronic kidney disease. Two studies involving 48 prevalent PD patients and 760 incident PD patients, respectively, evaluated the efficacy of PhA in predicting patient survival [10,11]. These studies showed a positive association between PhA and patient survival in both univariate and multivariate analyses. However, PhA was analyzed after dichromatic division in these studies. The cut-off values were 6.0° in Mushnick’s study and 4.95° in Huang’s study, based on their median values [10,11]. In addition, Reinaldi et al. evaluated whether PhA is an appropriate indicator for nutritional status; however, they were unable to conclude that PhA is an accurate independent indicator of malnutrition due to a high risk of bias in the index test [27]. Further studies regarding the significance of PhA for predicting clinical outcomes in PD patients are necessary. In our study, patients were divided into three groups regardless of specific cut-off values, and we observed better prognosis as the tertile decreased. Our results would be helpful in strengthening the efficacy of PhA in predicting patient survival in PD patients. Furthermore, our study suggests that PhA may be a more effective predictor than the other classical indicators, such as serum albumin, BMI, C-reactive protein, nPNA, or urine volume.

An association between PhA and technique failure was also an important result in our study. Shu et al. evaluated five studies using PD patients and assessed the impact of overhydration on technique failure in PD patients [28]. Volume overload can be associated with various pathologic conditions in PD patients, such as heart problems, peritonitis, malnutrition, or inflammation, which lead to increased technique failure. In our data, PhA is highly correlated with volume overload, with a correlation coefficient of 0.874 (*p* < 0.001, data were not shown) between PhA and extracellular water/total body water. A high correlation between the two variables would explain the association between PhA and technique failure in PD patients.

PhA is influenced by various factors, including age, sex, body mass index, malnutrition, hydration status, inflammation, displacement of body fluids (ascites or pleural effusion), and the proportion of muscle and fat mass [18,29]. Therefore, the accuracy of PhA measurements is generally accepted within healthy populations, with reference values of PhA varying based on sex, body size, and age within the same population. However, the accuracy of PhA measurements can be compromised in populations affected by various pathologies. Notably, PhA is particularly influenced by malnutrition. Therefore, monitoring changes in PhA alongside changes in serum albumin, a well-established nutritional status marker, can be valuable. In this study, we observed a decrease in serum albumin as the tertile of PhA decreased. Malnutrition often results in an increased extracellular water/intracellular water ratio and extracellular water/cell mass ratio [30,31,32], leading to a decrease in PhA through reduced reactance. A decrease in albumin levels is a significant indicator of malnutrition and contributes to changes in PhA. It is challenging to determine whether the decrease in PhA in these patients is solely an indirect result of malnutrition or if it holds independent significance. Moreover, factors such as inflammation or fluid overload can alter albumin levels, irrespective of nutritional status, consequently affecting PhA. Therefore, to accurately interpret changes in PhA that are independent of other factors, a comprehensive assessment of various indicators is essential. Currently, there is insufficient evidence to interpret the meaning of PhA changes solely based on independent measurements. To overcome this, it is necessary to establish diverse reference values for different condition groups.

To understand the independent impact of PhA amid these numerous influences, conducting subgroup analyses with a large sample size is recommended. In the actual dataset of our study, we observed significant differences in many baseline characteristics across PhA tertiles. Cox regression analyses also revealed a trend of decreasing HRs in multivariate survival analysis compared to univariate analysis. While univariate analysis showed statistical significance in cases of technique failure, the lack of significance upon adjusting for other variables can be partly attributed to the limited sample size. Nevertheless, it is evident that PhA was influenced by confounding factors such as age, volume status, cell mass, and comorbidities.

The complexity of various factors and concerns about accuracy pose challenges when applying PhA measurements to dialysis patients. These patients are often distinct from the general population owing to their health issues and lack tailored bioelectrical impedance analysis measurement reference values. Relying on values derived from the general population raises significant concerns. Additionally, the lack of adequate reimbursement for PhA measurements due to these influences, especially in countries like the Republic of Korea where systematic assessment for reimbursement has not been conducted, could limit widespread use. Moreover, accurate bioimpedance analysis measurements in PD patients require serial data and should be conducted under conditions that minimize the impact of various factors, such as infection or volume overloading. The timing of measurements, specifically after adequate redistribution between volume compartments following peritoneal dialysate drainage, can also act as a limiting factor for extensive use in PD patients.

This study has several limitations. Firstly, it was a single-center, retrospective study. Secondly, the limited sample size restricted the adjustment for confounding factors and subgroup analyses based on various characteristics. Thirdly, repeated PhA measurements were not included. Lastly, the influence of numerous confounding factors on PhA can complicate the assessment of its independent association with outcomes, even with multivariate or subgroup analyses outcomes. To address these limitations, prospective studies involving larger patient populations and data with repeated and follow-up measurements are essential.

## 5. Conclusions

The present study demonstrated that PhA may be an effective indicator for predicting patient or technique survival in PD patients. Furthermore, it suggests that the routine measurement of PhA and pre-emptive intervention to recover PhA according to causes of low PhA may help improve patient or technique survival in PD patients.

## Figures and Tables

**Figure 1 nutrients-15-04991-f001:**
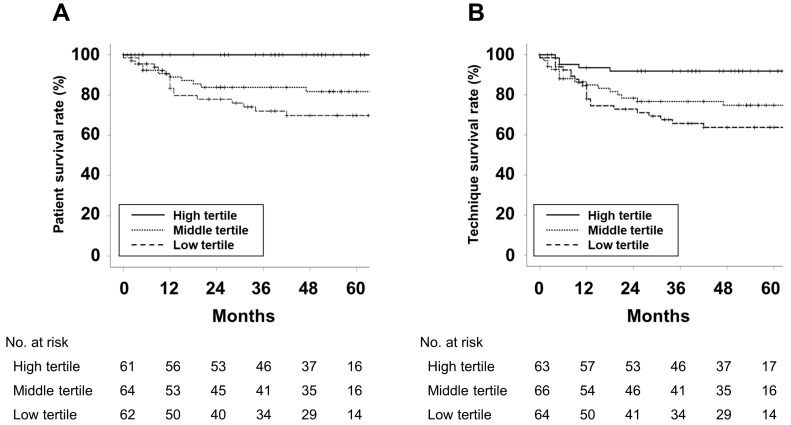
Kaplan–Meier curves for (**A**) patient survival and (**B**) technique survival. *p*-values were calculated using the Log-rank test. The 5 year patient survival rates for the high, middle, and low tertiles were 100%, 81.7%, 69.9%, respectively (*p* < 0.001). The 5 year technique survival rates for the high, middle, and low tertiles were 91.9%, 74.8%, 63.7%, respectively (*p* = 0.004).

**Figure 2 nutrients-15-04991-f002:**
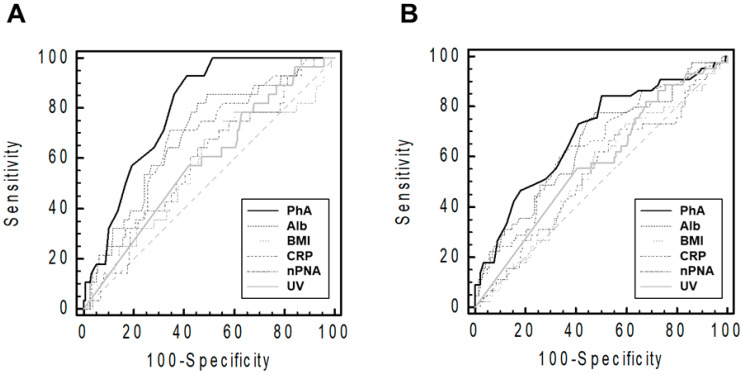
Receiver operating characteristic curves of the performance of various indicators in predicting patients’ deaths (**A**) and technique failures (**B**). The graph was plotted using raw PhA values. The sample size and median follow-up duration were 199 patients and 54 (19–88) months, respectively. Abbreviations: CRP, C-reactive protein; PhA, phase angle; Alb, serum albumin; BMI, body mass index; nPNA, normalized protein equivalent of total nitrogen appearance; UV, urine volume.

**Table 1 nutrients-15-04991-t001:** Clinical characteristics of participants according to tertiles of phase angle.

Characteristics	Low Tertile (*n* = 66)	Middle Tertile (*n* = 68)	High Tertile (*n* = 65)	*p*-Value
Age (years)	59.4 ± 11.7	57.3 ± 11.2	49.8 ± 11.9 ^a,b^	<0.001
Sex (male)	30 (45.5%)	35 (51.5%)	48 (73.8%)	0.003
Davies comorbidity index				<0.001
Low-risk group	19 (28.8%)	20 (29.4%)	40 (61.5%)	
Intermediate-risk group	37 (56.1%)	45 (66.2%)	24 (36.9%)	
High-risk group	10 (15.2%)	3 (4.4%)	1 (1.5%)	
Automated peritoneal dialysis (%)	14 (21.2%)	21 (30.9%)	22 (33.8%)	0.245
Dialysis vintage (months)	64 (37–108)	51 (26–80)	48 (25–86)	0.367
Body mass index (kg/m^2^)	23.9 (21.9–26.2)	23.9 (21.7–25.6)	24.8 (22.5–27.9) ^b^	0.025
Weekly Kt/V_urea_	1.93 ± 0.43	1.87 ± 0.43	1.96 ± 0.51	0.454
C-reactive protein (mg/dL)	0.14 (0.06–0.45)	0.18 (0.06–0.46)	0.17 (0.04–0.34)	0.519
Urine volume (mL/day)	0 (0–500)	0 (0–310)	355 (0–1200) ^a,b^	0.001
DP4Cr	0.69 ± 0.16	0.64 ± 0.11	0.65 ± 0.12	0.509
Phosphorus (mg/dL)	4.7 ± 1.4	5.0 ± 1.3	5.0 ± 1.5	0.597
Calcium (mg/dL)	8.3 ± 0.9	8.3 ± 1.0	8.3 ± 1.0	0.980
Potassium (mEq/L)	4.5 ± 0.8	4.6 ± 0.6	4.6 ± 0.6	0.432
Sodium (mEq/L)	137 (134–139)	136 (134–139)	137 (134–139)	0.615
Albumin (g/dL)	3.3 ± 0.5	3.6 ± 0.4 ^a^	3.8 ± 0.4 ^a,b^	<0.001
nPNA (g/kg/day)	0.78 ± 0.21	0.85 ± 0.23	0.88 ± 0.17 ^a^	0.028
Alkaline phosphatase (IU/L)	109 (86–148)	112 (76–148)	102 (76–135)	0.147
Intact parathyroid hormone (pg/mL)	269 (126–431)	285 (152–438)	314 (176–555)	0.235
Duration of follow-up (months)	41 (12–86)	55 (16–90)	59 (37–96)	0.241

Categorized data are summarized as frequency counts and percentages. Normally distributed continuous variables are summarized as mean ± standard deviation, while non-normally distributed continuous variables are summarized as median (interquartile range). Statistical significance among the three tertiles was assessed using the Kruskal–Wallis test for continuous variables that did not follow a normal distribution and one-way ANOVA for normally distributed continuous variables. Subsequently, Bonferroni’s post hoc test was performed to identify specific differences between the two groups for continuous variables. Categorical data were compared using Pearson’s chi-square or Fisher’s exact test, depending on the sample size. Abbreviations: DP4Cr, dialysate-to-creatinine concentration 4-h ratio; nPNA, normalized protein equivalent of total nitrogen appearance. Note: ^a^: compared with the low tertile, *p* < 0.05; ^b^: compared with the middle tertile, *p* < 0.05.

**Table 2 nutrients-15-04991-t002:** Cox regression analyses exploring the variables associated with patient and technique survival.

	Patient Survival				Technique Survival			
	Univariate		Multivariate		Univariate		Multivariate	
	HR (95% CI)	*p*	HR (95% CI)	*p*	HR (95% CI)	*p*	HR (95% CI)	*p*
Tertile of PhA (decrease 1 tertile)	3.25 (1.81–5.84)	<0.001	2.48 (1.32–4.66)	0.005	1.88 (1.28–2.77)	0.001	1.42 (0.92–2.17)	0.100
Age (ref: <65 years)	4.16 (2.00–8.62)	<0.001	3.60 (1.72–7.55)	0.001	2.60 (1.43–4.73)	0.002	2.31 (1.26–4.22)	0.007
Sex (ref: male)	1.71 (0.82–3.56)	0.150			1.15 (0.64–2.06)	0.640		
BMI (increased 1 kg/m^2^)	1.03 (0.94–1.12)	0.586			1.01 (0.94–1.09)	0.772		
UV (increase 1 mL/day)	1.00 (1.00–1.00)	0.112			1.00 (1.00–1.00)	0.158		
Albumin (increase 1 g/dL)	0.36 (0.19–0.69)	0.002	0.68 (0.34–1.37)	0.281	0.41 (0.24–0.70)	0.001	0.60 (0.33–1.06)	0.077
nPNA (increase 1 g/kg/day)	0.28 (0.05–1.77)	0.177			0.62 (0.15–2.65)	0.523		
Davies risk index (increase 1 grade)	2.45 (1.35–4.46)	0.003	2.01 (1.10–3.65)	0.023	1.95 (1.21–3.14)	0.006	1.63 (1.00–2.65)	0.051

Multivariate analysis was adjusted for variables with statistical significance in the univariate model. Abbreviations: HR, hazard ratio; CI, confidence interval; PhA, phase angle; BMI, body mass index; UV, urine volume; nPNA, normalized protein equivalent of total nitrogen appearance.

**Table 3 nutrients-15-04991-t003:** Performance analysis of multivariate models with and without PhA employing AUCs, IDI, and NRI.

Models	AUC	Difference between AUCs	Relative IDI		Category-Free NRI	
		Values	Values	*p*-Value	Values	*p*-Value
Patient death						
Multivariate model	0.79		–	–	–	–
Multivariate model with PhA	0.85	0.05	0.56	0.010	0.58	0.004
Technique failure						
Multivariate model	0.71		–	–	–	–
Multivariate model with PhA	0.73	0.02	0.19	0.051	0.28	0.090

The dependent variable was patient survival or technique survival at the end of the follow up period. The multivariate model included age, serum albumin, and Davies risk index, and the IDI and NRI analyses were performed using raw PhA values. Abbreviations: PhA, phase angle; AUC, area under curve; IDI, integrated discrimination improvement; NRI, net reclassification improvement.

## Data Availability

All data relevant to the study are included in the article.

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
