# Peer review of "Phase Angle as a Risk Factor for Mortality in Patients Undergoing Peritoneal Dialysis"

_nutrients, 2023, doi:10.3390/nu15234991_

Round 1

Reviewer 1 Report

Comments and Suggestions for Authors

Thank you for the opportunity to review this manuscript. Kang and Do report on a cohort study investigating the association between phase angle and survival in patients on peritoneal dialysis. Bioimpedance is an under-utilised tool and of interest to the renal community.

Comments:

1.      In the methods section, please explain why you have chosen to use IDI and NRI methods for analysis.

2.      In the methods section, for what model have you calculated an AUROC for? Please clearly specific the model.

3.      For lines 126-127 on page 3 and legend of Table 1, please specify for what characteristic the tertiles are based on

4.      In the discussion, please mention any shortcomings of using phase angle including other variables that can influence its measurement and any barriers to implementing phase angle measurements on a wider scale in more dialysis units.

5.      Please mention possible impact of confounding in the limitations section of the discussion.

6.      In Figure 2, please specify the sample size and duration of follow up so that Figure 2 can be more easily interpreted. Are these graphs based on raw phase angle measurement or phase angle tertile?

7.      On line 166 page 5 you have repeated the word cachexia: “cachexia for 1 (5.6%) for cachexia”

8.      In table 1, albumin was significantly higher from low to middle to high tertile. Can you hypothesise what is driving this trend?

9.      For table 1 do you have data on duration of follow up for each tertile?

10.   In table 2, the hazard ratios change after adjustment in the multivariate model. Can you please check for multicollinearity between covariates in the multivariate model.

11.   For table 2, have you considered using phase angle raw measurement instead of tertile in the cox regression analysis?

12.   Also in table 2, phase angle tertile no longer becomes significant for technique survival after adjusting for covariates and attenuates for patient survival – it is possible that the information captured in phase angle is also captured in age, albumin and davies risk index (all 3 variables seem significantly different between the three tertiles in table 1)?

13.   For table 3, please specify if phase angle tertile or measurement is being used.

14.   For figure 1, please indicate the number at risk and/or events underneath the Kaplan-meier plots. Please indicate in figure 1 legend that the p-value is calculated by log-rank test.

15.   For measuring phase angle, did the patients have a peritoneal dialysate dwell in their abdomen at the time of measurement or were they dry? If patient did or did not have a dwell would it affect the result and would the type of peritoneal dialysate fluid in the abdomen also affect the result?

Author Response

Reviewer 1

Thank you for the opportunity to review this manuscript. Kang and Do report on a cohort study investigating the association between phase angle and survival in patients on peritoneal dialysis. Bioimpedance is an under-utilised tool and of interest to the renal community.

Dear reviewer 1,

Thank you for your valuable feedback. We have addressed your comments and made improvement to our manuscript as follows:

Comments:

  1. In the methods section, please explain why you have chosen to use IDI and NRI methods for analysis.

Answer: We have explained our choice of using IDI and NRI methods for analysis. We mentioned that Pencina et al. proposed these to assess the improvement in predictive performance when a new factor is added to existing prediction models. Following their suggestion, we conducted IDI and NRI analyses to evaluate whether the inclusion of phase angle in a mortality prediction model based on existing risk factors enhances the predictive accuracy. We have added these comments in the Methods section.

  1. In the methods section, for what model have you calculated an AUROC for? Please clearly specific the model.

Answer: In the Methods section, we have specified that DeLong’s model was used to compare the AUROC.

  1. For lines 126-127 on page 3 and legend of Table 1, please specify for what characteristic the tertiles are based on

Answer: We have specified in the text and the legend of Table 1 that the tertiles are based on phase angle.

  1. In the discussion, please mention any shortcomings of using phase angle including other variables that can influence its measurement and any barriers to implementing phase angle measurements on a wider scale in more dialysis units.

Answer: We have added the following to the Discussion section: “PhA is influenced by various factors, including age, sex, body mass index, malnutrition, hydration status, inflammation, displacement of body fluids (ascites or pleural effusion), and the proportion of muscle and fat mass [1,2]. Therefore, the accuracy of PhA measurements is generally accepted within healthy populations, with reference values of PhA varying based on sex, body size, and age within the same population. However, the accuracy of PhA measurements can be compromised in populations affected by various pathologies. Notably, PhA is particularly influenced by malnutrition. Therefore, monitoring changes in PhA alongside changes in serum albumin, a well-established nutritional status marker, can be valuable. In this study, we observed a decrease in serum albumin as the tertile of PhA decreased. Malnutrition often results in an increased extracellular water/intracellular water ratio and extracellular water/cell mass ratio [3-5], leading to a decrease in PhA through reduced reactance. This decrease in albumin levels is a significant indicator of malnutrition and contributes to changes in PhA. It is challenging to determine whether the decrease in PhA in these patients is solely an indirect result of malnutrition or if it holds independent significance. Moreover, factors such as inflammation or fluid overload can alter albumin levels, irrespective of nutritional status, consequently affecting PhA. Therefore, to accurately interpret changes in PhA that are independent of other factors, a comprehensive assessment of various indicators is essential. Currently, there is insufficient evidence to interpret the meaning of PhA changes solely based on independent measurements. To overcome this, it is necessary to establish diverse reference values for different condition groups.

To understand the independent impact of PhA amid these numerous influences, conducting subgroup analyses with a large sample size is recommended. In the actual dataset of our study, we observed significant differences in many baseline characteristics across PhA tertiles. Cox regression analyses also revealed a trend of decreasing HRs in multivariate survival analysis compared to univariate analysis. While univariate analysis showed statistical significance in cases of technique failure, the lack of significance upon adjusting for other variables can be partly attributed to the limited sample size. Nevertheless, it is evident that PhA was influenced by confounding factors such as age, volume status, cell mass, and comorbidities.

The complexity of various factors and concerns about accuracy pose challenges when applying PhA measurements to dialysis patients. These patients, often distinct from the general population owing to their health issues, lack tailored bioelectrical impedance analysis measurement reference values. Relying on values derived from the general population raises significant concerns. Additionally, the lack of adequate reimbursement for PhA measurements due to these influences, especially in countries like the Republic of Korea, where systematic assessment for reimbursement has not been conducted, could limit widespread use. Moreover, accurate bioimpedance analysis measurements in PD patients require serial data and should be conducted under conditions that minimize the impact of various factors, such as infection or volume overloading. The timing of measurements, specifically after adequate redistribution between volume compartments following peritoneal dialysate drainage, can also act as a limiting factor for extensive use in PD patients.”

We have included the following references:

[1] Norman, K.; Stobäus, N.; Pirlich, M.; Bosy-Westphal, A. Bioelectrical phase angle and impedance vector analysis--clinical relevance and applicability of impedance parameters. Clin Nutr. 2012, 31(6), 854-861.

[2] Bellido, D.; García-García, C.; Talluri, A.; Lukaski, H.C.; García-Almeida, J.M. Future lines of research on phase angle: Strengths and limitations. Rev Endocr Metab Disord. 2023, 24(3), 563-583.

[3] Shizgal, H.M. The effect of malnutrition on body composition. Surg Gynecol Obstet. 1981, 152(1), 22-26.

[4] Selberg, O.; Selberg, D. Norms and correlates of bioimpedance phase angle in healthy human subjects, hospitalized patients, and patients with liver cirrhosis. Eur J Appl Physiol. 2002, 86(6), 509-516.

[5] Dumler, F.; Kilates, C. Body composition analysis by bioelectrical impedance in chronic maintenance dialysis patients: comparisons to the National Health and Nutrition Examination Survey III. J Ren Nutr. 2003, 13(2), 166-172. 

  1. Please mention possible impact of confounding in the limitations section of the discussion.

Answer: We have added the following to the Limitation section of Discussion: “Lastly, the influence of numerous confounding factors on PhA can complicate the assessment of its independent association with outcomes, even with multivariate or subgroup analyses outcomes. To address these limitations, prospective studies involving larger patient populations and data with repeated and follow-up measurements are essential.”

Please refer to our answer to previous question.

  1. In Figure 2, please specify the sample size and duration of follow up so that Figure 2 can be more easily interpreted. Are these graphs based on raw phase angle measurement or phase angle tertile?

Answer: We have provided additional information in Figure 2, specifying the sample size (199 patients) and median follow-up period (54 [1988] months). Additionally, we have indicated that the AUROC for phase angle was calculated using raw phase angle values in the figure legend.

  1. On line 166 page 5 you have repeated the word cachexia: “cachexia for 1 (5.6%) for cachexia”

Answer: Thank you for your meticulous review. We have removed the redundant term, “for cachexia”.

  1. In table 1, albumin was significantly higher from low to middle to high tertile. Can you hypothesise what is driving this trend?

Answer: We have added the following to the Discussion section: “In this study, we observed a decrease in serum albumin as the tertile of phase angle decreased. Malnutrition often results in an increased extracellular water/intracellular water ratio and extracellular water/cell mass ratio, leading to a decrease in phase angle through reduced reactance. This decrease in albumin levels is a significant indicator of malnutrition and contributes to change in phase angle.

We have included the following references:

[1] Shizgal HM. The effect of malnutrition on body composition. Surg Gynecol Obstet. 1981 Jan;152(1):22-6.

[2] Selberg O, Selberg D. Norms and correlates of bioimpedance phase angle in healthy human subjects, hospitalized patients, and patients with liver cirrhosis. Eur J Appl Physiol. 2002 Apr;86(6):509-16.

[2] Dumler F, Kilates C. Body composition analysis by bioelectrical impedance in chronic maintenance dialysis patients: comparisons to the National Health and Nutrition Examination Survey III. J Ren Nutr. 2003 Apr;13(2):166-72. 

  1. For table 1 do you have data on duration of follow up for each tertile?

Answer: The median duration of follow-up in low, middle, and high tertiles of phase angle were 41 (1286), 55 (1690), and 59 (3796) months, respectively. We have added this data to the Results section.

  1. In table 2, the hazard ratios change after adjustment in the multivariate model. Can you please check for multicollinearity between covariates in the multivariate model.

Answer: We have checked variance inflation factors (VIF) using regression analysis with variables of multivariate analysis (patient death as dependent variable and phase angle, age, serum albumin, and Davies risk index as independent variables). VIF was 1.16, 1.18, 1.35, and 1.18 for phase angle, age, serum albumin and Davies risk index, respectively. This indicated no multicollinearity among independent variables of multivariate analyses. We have added this information to the Methods and Results sections.

  1. For table 2, have you considered using phase angle raw measurement instead of tertile in the cox regression analysis?

Answer: Thank you for your suggestions. We have performed Cox regression analysis using raw phase angle values, our findings have been included in the Results section: “Furthermore, we performed Cox regression analysis using raw PhA values. In univariate analysis, the HR (95% confidence interval [CI]) for patient mortality was 0.34 (0.22–0.53, P < 0.001) for every 1° increase in PhA value. In multivariate analysis adjusted for age, serum albumin, and Davies risk index, the HR was 0.40 (0.24–0.68, P = 0.001) for every 1° increase in PhA value. Additionally, in univariate analysis, the HR (95% CI) for technique failure was 0.53 (0.38–0.75, P < 0.001) for every 1° increase in PhA value. In multivariate analysis adjusted for age, serum albumin, and Davies risk index, the HR was 0.68 (0.46–1.01, P = 0.058) for every 1° increase in PhA value.”

  1. Also in table 2, phase angle tertile no longer becomes significant for technique survival after adjusting for covariates and attenuates for patient survival – it is possible that the information captured in phase angle is also captured in age, albumin and davies risk index (all 3 variables seem significantly different between the three tertiles in table 1)?

Answer: We have added discussions in manuscript regarding this issue. Please refer to our answer to previous question.

  1. For table 3, please specify if phase angle tertile or measurement is being used.

Answer: We have indicated that raw phase angle values were used for IDI and NRI analyses in the footnote of Table 3.

  1. For figure 1, please indicate the number at risk and/or events underneath the Kaplan-meier plots. Please indicate in figure 1 legend that the p-value is calculated by log-rank test.

Answer: In Figure 1, we have added the number at risk for the three groups and indicated that the p-value was calculated using the log-rank test in the figure legend.

  1. For measuring phase angle, did the patients have a peritoneal dialysate dwell in their abdomen at the time of measurement or were they dry? If patient did or did not have a dwell would it affect the result and would the type of peritoneal dialysate fluid in the abdomen also affect the result?

Answer: We have clarified in text that phase angle measurements were performed after emptying the abdomen to exclude the effect of peritoneal dialysate. Notably, the presence of peritoneal dialysate is associated with an underestimation of PhA [1]. To account for this effect, all PhA measurements were conducted after the abdomen had been emptied.

We have added the following reference:

[1] Arroyo, D.; Panizo, N; Abad, S.; Vega, A.; Rincón, A.; de José, A.P.; López-Gómez, J.M. Intraperitoneal fluid overestimates hydration status assessment by bioimpedance spectroscopy. Perit Dial Int. 2015, 35(1), 85-89.

Thank you for your careful review, and we believe these changes enhance the clarity and completeness of our manuscript.

Reviewer 2 Report

Comments and Suggestions for Authors

The bioelectrical impedance analysis (BIA), particularly as characterized by the phase angle (PhA), is a promising method for medical diagnosis due to the ease of data acquisition. The PhA is increasingly recognized as a prognostic indicator in various medical conditions. This work can be a useful addition to the list.

However, I believe that it would be better to acknowledge the known variances and susceptibility to bias in the PhA. Previous works generally associate higher PhA with better cellular integrity and health prognosis. But as the authors note in the Discussion, this work finds the opposite trend. Another point that the authors might have highlighted is their finding that albumin is indeed shown as a significant biomarker. Rather than toning down its significance, it would be better to acknowledge its value and compare PhA with it. I think this work can be accepted after a minimal rephrasing of the descriptions regarding the accuracy and significance of PhA, since these caveats are already accepted in the field and do not compromise the integrity of the work.

Minor comments:

Line 8: phase angle (PhA): Please define PhA briefly, or at least specify that it relates to bioimpedance analysis, as the term 'phase angle' is used in various contexts across different fields.

Line 47: ...PhA is well-known as a precise indicator without ... -> consider rephrasing to: ...PhA is increasingly recognized as a prognostic indicator without ...

Line 240: Furthermore, our study showed that PhA is a better predictor than ... -> consider rephrasing: Furthermore, our study suggests that PhA may be a more effective predictor than ...

Comments on the Quality of English Language

I can detect a few minor stylistic errors that can be corrected and improved for better readability. For example,

Line 12: The PhA was measured using a multifrequency ... -> The PhA was measured using multifrequency ... (without 'a')

Author Response

Dear reviewer 2,

Thank you for your valuable feedback. We have addressed your comments and made improvement to our manuscript as follows:

Reviewer 2

The bioelectrical impedance analysis (BIA), particularly as characterized by the phase angle (PhA), is a promising method for medical diagnosis due to the ease of data acquisition. The PhA is increasingly recognized as a prognostic indicator in various medical conditions. This work can be a useful addition to the list.

However, I believe that it would be better to acknowledge the known variances and susceptibility to bias in the PhA. Previous works generally associate higher PhA with better cellular integrity and health prognosis. But as the authors note in the Discussion, this work finds the opposite trend. Another point that the authors might have highlighted is their finding that albumin is indeed shown as a significant biomarker. Rather than toning down its significance, it would be better to acknowledge its value and compare PhA with it. I think this work can be accepted after a minimal rephrasing of the descriptions regarding the accuracy and significance of PhA, since these caveats are already accepted in the field and do not compromise the integrity of the work.

Answer: We have appreciated your insightful comments and feedback on our manuscript. We have included the following in the Discussion section: “PhA is influenced by various factors, including age, sex, body mass index, malnutrition, hydration status, inflammation, displacement of body fluids (ascites or pleural effusion), and proportions of muscle mass or fat mass [1,2]. Therefore, the accuracy of PhA is generally accepted within healthy populations, with reference of PhA varying based on sex, body size, or age within the same population. However, the accuracy of PhA measurements can be compromised in populations affected by various pathologies. Notably, PhA is particularly influenced by malnutrition. Therefore, monitoring changes in PhA alongside changes in serum albumin, a well-established nutritional status marker, can be valuable. In this study, we observed a decrease in serum albumin as the tertile of PhA decreased. Malnutrition often results in an increased extracellular water/intracellular water ratio and extracellular water/cell mass ratio, leading to a decrease in PhA through reduced reactance. This decrease in albumin levels is a significant indicator of malnutrition and contributes to changes in PhA. It is challenging to determine whether the decrease in PhA in these patients is solely an indirect result of malnutrition or if it holds independent significance. Moreover, factors such as inflammation or fluid overload can alter albumin levels, irrespectively of nutritional status, consequently affecting PhA. Therefore, to accurately interpret changes in PhA that are independent of other factors, a comprehensive assessment of various indicators is essential. Currently, there is insufficient evidence to interpret the meaning of PhA changes solely based on independent measurements. To overcome this, it is necessary to establish diverse reference values for different condition groups.”

We have included the following references:

[1] Norman K, Stobäus N, Pirlich M, Bosy-Westphal A. Bioelectrical phase angle and impedance vector analysis--clinical relevance and applicability of impedance parameters. Clin Nutr. 2012 Dec;31(6):854-61.

[2] Bellido D, García-García C, Talluri A, Lukaski HC, García-Almeida JM. Future lines of research on phase angle: Strengths and limitations. Rev Endocr Metab Disord. 2023 Jun;24(3):563-583.

Minor comments:

Line 8: phase angle (PhA): Please define PhA briefly, or at least specify that it relates to bioimpedance analysis, as the term 'phase angle' is used in various contexts across different fields.

Answer: We have included the following sentence in the Abstract: “Phase angle (PhA) is measured using bioimpedance analysis and calculated using body reactance and resistance in the waveform at 50 kHz.”

Line 47: ...PhA is well-known as a precise indicator without ... -> consider rephrasing to: ...PhA is increasingly recognized as a prognostic indicator without ...

Answer: We have revised the relevant sentence as follows: “Therefore, PhA is increasingly recognized as a prognostic indicator without transformation from various equations.”

Line 240: Furthermore, our study showed that PhA is a better predictor than ... -> consider rephrasing: Furthermore, our study suggests that PhA may be a more effective predictor than ...

Answer: We have revised the relevant sentence as follows: “Furthermore, our study suggests that PhA may be a more effective predictor than the other classical indicators, such as serum albumin, BMI, C-reactive protein, nPNA, or urine volume.”

Comments on the Quality of English Language

I can detect a few minor stylistic errors that can be corrected and improved for better readability. For example,

Line 12: The PhA was measured using a multifrequency ... -> The PhA was measured using multifrequency ... (without 'a')

Answer: Thank you for pointing these stylistic errors out. We have revised the relevant sentence as follows: “The PhA was measured using multifrequency bioimpedance analysis.”

We have performed thorough editing with the assistance of two native English editors to enhance the overall readability and fluency of the manuscript. We have attached a certification file for English edition.

Once again, we thank you for your valuable comments and suggestions, which have greatly contributed to the improvement of our manuscript.
